# Creatinine as a Urinary Marker of the Purine Derivatives Excretion in Urine Spot Samples of Lambs Fed Peach Palm Meal

**DOI:** 10.3390/ani12091195

**Published:** 2022-05-06

**Authors:** Taiala Cristina de Jesus Pereira, Mara Lúcia Albuquerque Pereira, Gleidson Giordano Pinto de Carvalho, Herymá Giovane de Oliveira Silva, Alana Batista dos Santos, Douglas dos Santos Pina, Leandro Borges Sousa

**Affiliations:** 1Post-Graduate Program in Animal Science, State University of Southwest Bahia, BR 415, Km 03, Itapetinga 45700-000, Brazil; marauesb@yahoo.com.br (M.L.A.P.); heryma@gmail.com (H.G.d.O.S.); alanasantos10@hotmail.com (A.B.d.S.); leandroborgessousa@hotmail.com (L.B.S.); 2Department of Animal Science, Federal University of Bahia, Ademar de Barros Avenue, 500, Ondina, Salvador 40170-110, Brazil; gleidsongiordano@yahoo.com.br (G.G.P.d.C.); douglaspinaufba@gmail.com (D.d.S.P.)

**Keywords:** creatinine clearance, *Bactris gasipaes*, microbial synthesis, renal activity, total urine

## Abstract

**Simple Summary:**

Experimental trials designed to predict the microbial protein synthesis in the rumen via urinary purine derivatives require access to 24-h urinary volumes. This technique is non-invasive but requires access to 24-h urinary volumes, which is impracticable in grazing animals or a large number of feedlot sheep. The challenge is to provide the possibility of using the purine derivatives and creatinine concentrations in a spot urine sample and the daily creatinine excretion in urine as accurate estimators of purine derivatives’ excretion per day.

**Abstract:**

The objective was to evaluate the influence of diets on lambs using different levels of peach palm meal as a replacement for maize (0, 10, 40, 60, and 85% of diet dry matter) on the endogenous creatinine clearance (CC), urine concentration ratio of purine derivatives to creatinine (PDC index), and daily creatinine excretion (DCE) as a marker to estimate purine derivatives (PD) excretion from urinary spot samples collected at different time points (4, 8, 12, 16, 20, 24 h after morning feeding) compared to 24-h total urine collection. The measured parameters were voluntary intake, urinary volume, CC, DCE, the concentration of plasma creatinine, and PD and purine derivatives’ excretion (PDE). Five lambs were allocated to metabolic cages and distributed in a 5 × 5 Latin square. Urine collection was taken daily on days 16 to 19 of each experimental period. The inclusion of peach palm meal linearly reduced the intake of dry matter (g kg BW^−0.75^, *p =* 0.005), crude protein (g kg BW^−0.75^, *p =* 0.010), metabolizable energy (MJ kg BW^−0.75^, *p* = 0.010) and CC (*p <* 0.0001). It also quadratically affected the urinary volume (*p* = 0.008) and DCE (*p* = 0.004). There was a linear decrease for PDC index (*p* = 0.032) and PDE (*p* < 0.0001) measured in the 24-h total urine with peach palm meal levels. The different times of spot urine sampling did not affect (*p* > 0.05) the PDC index and PDE. Peach palm meal decreases the CC thereby compromising the use of a mean value of DCE as a PDE marker in spot urine samples. There is greater accuracy when using different values of DCE obtained for each diet as markers for the PDE in spot urine samples. Unconventional foodstuffs of low palatability affecting the voluntary intake of feed change the renal function.

## 1. Introduction

Peach palm (*Bactris gasipaes* Kunth) is native to the Amazonian region and is adapted to a wide range of ecological conditions in the humid tropical regions, it has become an agricultural species in the humid tropical regions of Brazil. The heart-of-palm industry directs part of the cultivation area to the production of fruit for the extraction of seeds. Hence, this process generates large amounts of fruit pulp waste, which does not yet have a suitable destination. The fruit pulp is a waste that could be used for the production of peach palm meal as an alternative feedstuff to maize for ruminants.

However, peach palm meal was found to have low palatability owing to the bitter taste of phenolic compounds and/or rancidness of the rich-unsaturated fatty acid lipid fraction, with consequences for the feed consumption and intake behavior [1,2,3,4,5,6].

There are a host of appetite and satiety mediators with interactive control of food voluntary intake, energy homeostasis, and body weight [7,8,9,10,11,12,13,14,15,16]. The neuropeptide Y (NPY) has been strongly implicated in the stimulation of feeding and also modulates bitter stimulation in the taste buds [11,13].

Activation of NPY receptors in the brain increases feeding intake and may affect kidney function, causing a vasoconstrictor effect on the renal vasculature and a decrease in renal blood flow [17,18,19,20,21,22,23]. Renal activity in animals can be evaluated with the use of clearance methods, in which endogenous creatinine can be used as a testing substance to determine the level of glomerular filtration [23,24,25].

The urinary purine derivatives’ excretion is an accurate estimator of microbial protein synthesized in the rumen and is easy to determine since it overcomes the disadvantages of more direct methods [26,27,28,29]. However, the purine derivatives’ excretion technique requires the quantitative collection of urine, which is not applicable in grazing animals or in a large group of animals.

An alternative to 24-h total urine collections is the possibility of using the concentration ratio of purine derivatives (PD) to creatinine (PDC index) in spot urine samples and the daily creatinine excretion as estimators of the PD excretion [27,30,31,32]. The spot urine collection technique to obtain the urinary excretion of PD is based on the principle that daily creatinine excretion does not vary with diet and only a sample of urine collected at 4 h after morning feeding is suitable [33,34].

Hence, we hypothesized that the PDC index and a single value of daily creatinine excretion as markers of the urinary output were not suitable for estimating the purine derivatives’ excretion in a spot urine sample when diets change the creatinine clearance. To test this hypothesis, this study aimed to evaluate the influence of different levels of peach palm meal as a replacement for maize (0, 10, 40, 60, and 85% of diet dry matter) on the endogenous creatinine clearance, daily creatinine excretion, concentrations of plasma creatinine, and the PD and PDC index. In addition, the accuracy of spot urine sampling was also assessed at different time points (4, 8, 12, 16, 20, 24 h after morning feeding) as estimators of purine derivatives’ excretion by lambs, compared to 24-h total urine collection.

## 2. Materials and Methods

### 2.1. Animal Care

The Ethics Commission of the State University of Southwest Bahia (UESB), Itapetinga Campus, protocol 11-2012, approved the experimental procedures of this study. This experiment was conducted in the sheep farming sector of the UESB, Itapetinga Campus, Bahia State, Brazil.

### 2.2. Experimental Design

Five crossbred (Santa Inês × undefined breed) lambs, intact males, with an approximate age of four months and body weight (BW) at the beginning of the experiment of 17.9 ± 2.0 kg were used. The animals were numbered, dewormed, and allocated in 1.0 × 0.8 m (0.8 m^2^) metabolic cages, randomly distributed in a 5 × 5 Latin square design. The experiment lasted 95 days, consisting of periods of 19 days each (14 days were used for adaptation to the diet and 5 days were used for sample collection).

### 2.3. Diet and Feeding

Diets were formulated with average crude protein (CP) at 139 g kg^−1^ and with average metabolizable energy (ME) at 12.26 MJ kg^−1^ on a dry matter (DM), formulated to allow a body weight gain rate of 250 gd^−1^ as recommended for lambs by the National Research Council [35]. The tested diets consisted of concentrate at 700 g kg^−1^ of diet DM composed with peach palm meal as a replacement for maize (0, 10, 40, 60, and 85% of diet DM). The concentrate was mixed with Tifton 85 hay (particle size at 5 cm) at 300 g kg^−1^ of diet DM at the time of diet supply.

The pulp of the pitted fruit (pericarp and mesocarp) was supplied by Indústria de Alimentos no Mercado de Palmitos (INACERES), located in Uruçuca-BA. The peach palm meal was produced in a flour mill at Instituto Federal Baiano (IFBAIANO), Uruçuca Campus-BA. The obtained pulp was dried in the sun for three consecutive days, with the material being turned over three times a day until its moisture content was reduced by half. Subsequently, it was disintegrated in a cassava grinder, and then the groundmass was roasted in a mechanized flour roaster. This roasting procedure lasted 30 to 40 min, with the mass being turned over using wooden squeegees until its final drying, at approximately 13 g kg^−1^ moisture.

Table 1 shows the proportions of ingredients and average nutrient contents of the Tifton 85 hay, peach palm meal, and diets, respectively. The diets (concentrate and Tifton 85 hay) were supplied for ad libitum intake, twice a day at 07:00 h and 16:00 h, to allow residual feed of 10%. The animals had free access to the water that was supplied in drinking troughs that were cleaned daily.

### 2.4. Collection and Laboratory Analyses

During each experimental period, on days 15 to 19, concentrates, Tifton 85 hay, and residual feed samples were taken. The intake of each animal was measured from the 15th to 19th day of each experimental period, calculated as the difference between the supplied feed (concentrate, Tifton 85 hay) and the residual feed. All samples were placed in plastic bags and frozen (−20 °C) for later analyses.

Samples of supplied and residual feed were analyzed for DM (method INCT—CA G—003/1), and total nitrogen (N) [36]. Concentrates and Tifton 85 hay supplied were analyzed for ash (method INCT-CA M-001/1), crude protein (CP; method INCT-CA N-001/1), and ether extract (EE; method INCT-CA G-004/1), according to [36]. For neutral detergent fiber (NDF) analysis, samples were treated with thermostable alpha-amylase, without the use of sodium sulfite and corrected for residual ash [37]. The NDF correction for nitrogen compounds and estimates of the neutral (NDIN) and acid (ADIN) detergent insoluble nitrogen compounds were carried out, according to [38]. Lignin (method INCT—CA F—005/1) was obtained based on the methodology described by [39], with the acid detergent fiber (ADF) residue treated with 72% sulfuric acid. The concentration of non-fiber carbohydrates (NFC) was calculated by adapting the method proposed by [40], utilizing NDF corrected for ash and protein [41]. The TDN content was calculated according to [42], but using NDF and NFC corrected for ash and protein, as shown in the equation: TDN (g kg^−1^) = _D_CP + _D_NDF + _D_NFC + 2.25 _D_EE, where: _D_CP = digestible crude protein; _D_NDF = digestible neutral detergent fiber; _D_NFC = digestible non-fiber carbohydrates; and _D_EE = digestible ether extract.

Animals were weighed at the beginning and end of each experimental period to obtain the mean BW of the respective experimental period. The total collection of urine (24 h) from each animal was held on the 16th to 19th day of each experimental period. Urine was collected using collecting funnels, which were attached to the animals and coupled with leading hoses to conduct urine to plastic containers with 100 mL of H_2_SO_4_ 20% (*v/v*) [43,44]. This solution was added to keep the urine pH below 4.0 [45], which was monitored during all the collection periods. At the end of each day, the urine pool was weighed, homogenized, filtered through cheesecloth layers, and a subsample (50 mL) was sampled and stored at −20 °C for further analyses.

On the second day of collection in each experimental period, i.e., on day 2, spot urine samples were collected at 4-h intervals over a period of 24 h. In this case, the urine sample was collected directly into Falcon^TM^ tubes attached to the animal every 4-h interval. Aliquots of 10 mL (spot urine) were diluted in 40 mL of H_2_SO_4_ 0.018 mol L^−1^, labelled appropriately, and stored at −20 °C for later analyses. The sum of the volume of each collected urine sample (10 mL) was used to obtain the total urine volume on day 2.

Four hours after the morning feeding, on day 2 of the total urine collection, blood samples were taken via puncture of the jugular vein, using 4 mL EDTA K2 Vacutainer^®^ tubes (Becton Dickinson Vacutainer System, Rutherford, NJ, USA). The samples were immediately centrifuged at 2200 g (Centrifuge himac CF-16RX II, Rotor type T15A36, Hitachi Koki) for 10 min to obtain blood plasma, which was analyzed for creatinine and stored at −20 °C for later analysis for urea, allantoin, xanthine–hypoxanthine, and uric acid. The concentrations of creatinine and uric acid in urine and in blood plasma were determined using commercial kits from Bioclin^®^ (ref. K016 and ref. K139, Delft, the Netherlands). The urinary and plasma concentrations of allantoin and xanthine–hypoxanthine were determined by colorimetric methods, as specified by [26]. The sum of the urinary excretions of allantoin, xanthine–hypoxanthine, and uric acid was used to obtain the PDE.

### 2.5. Calculations

For testing the effect of the number of days of urine collection, the combinations were realized by the mathematical calculation using the observed values on day 1 = T_1_ sample; day 2 = (T_1_ + T_2_)/2; day 3 = (T_1_ + T_2_ + T_3_)/3; and day 4 = (T_1_ + T_2_ + T_3_ + T_4_)/4, for comparison among the days of total collection (24, 48, 72, and 96 h).

The endogenous creatinine clearance (CC) was calculated, as described by [24]:(1)$$\mathrm{CC}=\frac{C}{\mathrm{PC}}\times \frac{\mathrm{VU}\;}{1440}$$
where: C = the concentration of 24-h total urine creatinine (mg dl^−1^); PC = the concentration of plasma creatinine (mg dl^−1^); and VU = the urine volume (mL) per min.

The urinary volumes estimated from the spot urine samples were calculated as:(2)$$\mathrm{VU}=\frac{\mathrm{DCE}}{C}$$
where: VU = estimated urine volume (L); DCE = the mean daily creatinine excretion (mg kg^−1^ BW); and C = the concentration in the spot urine sample (mg L^−1^).

We used the single mean value of DCE (28.12 mg kg^−1^ BW) in the evaluation of urinary spot samples collected at different time points (4, 8, 12, 16, 20, 24 h after morning feeding) to obtain PDE in lambs and to compare with 24-h total urine collection.

The different mean values of DCE obtained in each diet (30.5, 27.9, 25.9, 27.8, and 28.4 mg kg^−1^ BW) from 24-h total urine collection were utilized to estimate the PDE in a spot urine sample collected 4 h after morning feeding, to evaluate the different peach palm meal levels.

The PDC index was calculated according to [30]:(3)$$\mathrm{PDC}\;\mathrm{index}=\frac{\mathrm{PD}\;}{C}\times {\mathrm{BW}}^{0.75}$$
where: PD = the purine derivatives concentration (mmol L^−1^); C = the urinary creatinine concentration (mmol L^−1^); and BW^0.75^ = the metabolic body weight (kg).

### 2.6. Statistical Analyses

The data were statistically analyzed using the MIXED MODEL procedure [46]. In the 5 × 5 design structure Akaike’s Information Criteria (AIC) was used to select the appropriate covariance structure using Compound symmetry (TYPE = CS). For exploratory data analysis the Shapiro–Wilk test was used for normal distribution and the Bartlett test was used for homogeneity of variance.

Data of urine volume (measured), creatinine clearance (CC), the plasma concentration of creatinine and purine derivatives, daily creatinine excretion (DCE), purine derivatives excretion (PDE), and PDC index were analyzed as a Latin square with lambs fed five diets (D: peach palm meal levels replacing maize) and five experimental periods (P). These variables were studied on four different days (C: days 1, 2, 3, and 4 of 24-h total urine collections) as repeated measures. The main effects of D (diet) and C (collection days) were included as fixed effects and animals (A) and experimental periods (P) as random variables.

Estimated data of urine volume and the PDE and PDC index were analyzed according to the description above, using day 1 of the 24-h total urine collection (C) and time points of spot urine sampling at 4-h intervals after morning feeding (T) in lambs fed diets (D) containing peach palm meal replacing maize.

Nutrients’ intake and urinary excretion of purine derivatives and PDC index using spot sample collected at 4 h after morning feeding in lambs fed diets (D) containing peach palm meal replacing maize were analyzed according to the model:(4)$$Y_{\mathrm{ij}(k)}=\;\mu +P_{i}+A_{j}+D_{\left(k\right)+}\varepsilon_{\mathrm{ij}(k)\;i,j,k\;=1,\ldots ,r}$$
where: Y_ij__(k)_ = intake, plasma concentration and urinary excretion of creatinine and PD in evaluation k, animal j and period i; μ = overall mean; P_i_ = effect of period i; A_j_ = effect of animal j; D_(k)_ = fixed effect of diet k; treatment effect, ε_ij__(k)_ = random error with mean 0 and variance σ^2^; r = number of diets, periods and animals.

The evaluation of the effects of the level of maize replacement by peach palm meal (D) and collection time of spot urine at 4-h intervals (T) was performed by polynomial contrast and regression analysis, in case of significant effects on ANOVA. For diets (0, 10, 40, 60, and 85% on diet dry matter) the ORPOL function in PROC IML was used to obtain the appropriate coefficients for the CONTRAST statement. In Regression, the ARH (1) correlation structure was used. The general cubic model was adjusted and then phased out to construct the final model. The model for simple linear regression was:(5)$$Y_{i\;={\;\beta }_{0}}+\beta_{1x_{1}}+\varepsilon_{i\;\;\;i\;=1,\ldots ,\;n}$$
where: Y_i_ = observation i of dependent variable y; β_0_, β_1_ = regression parameters; x_i_ = observation i of independent variable × (diets or time); ε_i_ = random error. Model assumptions: E(ε_i_) = 0, mean of errors is equal to zero; Var(ε_i_) = σ^2^, variance is constant for every ε_i_, that is, variance is homogeneous; Cov(ε_i_, ε_i_’) = 0, i ≠ i’, errors are independent, the covariance between them is zero; usually, it is assumed that εi are normally distributed, ε_i_ ~ N (0, σ^2^). When that assumption is met the regression model is said to be normal.

Results from treatments are presented as least square means which were compared by contrast with *p <* 0.05. For ANOVA, the critical level adopted was 0.05 *< p <* 0.10 for type I error.

## 3. Results

The inclusion of peach palm meal replacing maize in the diets of lambs linearly reduced the daily intake (g kg BW^−0.75^) of DM (*p =* 0.005), CP (*p =* 0.010), metabolizable energy (ME, *p =* 0.010), and the ether extract (EE) intake was not affected (*p =* 0.663) (Table 2).

The diets affected the urine volume (*p =* 0.008), such that the minimum urine volume was observed to the level of 42% of peach palm meal replacing maize. The creatinine clearance reduced linearly (*p <* 0.0001); the plasma concentrations of creatinine increased (*p =* 0.011) by increasing the levels of peach palm meal in diets (Table 3). The daily creatinine excretion was also altered (*p =* 0.016) by the peach palm meal levels in the diets and the quadratic equation estimated the minimum point at 50% of peach palm meal replacing maize (Table 3).

The purine derivatives’ excretion in urine decreased linearly (*p* < 0.0001), PD in plasma did not change (*p =* 0.818), and the PDC index showed a linear decrease (*p =* 0.034) with the peach palm meal levels.

For the urine volume, purine derivatives’ excretion, and PDC index, differences associated with the experimental diets were found in the spot urine samples collected at 4-h intervals after the morning feeding (Table 4). It is consistent with the results obtained by the collection of 24-h total urine (Table 3). In contrast, there was an effect (*p =* 0.0002 and *p =* 0.004) with a linear component (*p =* 0.006 and *p =* 0.015) and a cubic effect (*p =* 0.0003 and *p =* 0.002) of the peach palm meal levels on the respective purine derivatives’ excretion and the PDC index. Therefore, spot urine sampling was less accurate to explain the variation of purine derivatives’ excretion depending on the diet, provided the urine volume was not properly estimated (Table 4). Hence, the cubic effect shows other possible interfering factors, for example, renal activity.

The effect of diet on the PDC index obtained in the spot urine samples at different time points after morning feeding was inconsistent with the results for the PDC index obtained in the 24-h total urine samples and with spot urine collected at a time point of 4 h after morning feeding (Table 3, Table 4 and Table 5).

The time of spot urine sampling (4, 8, 12, 16, 20, 24 h) did not affect the urine volume urine (*p* = 0.314) estimated by an average value from all diets (28.12 mg kg^−1^ BW), purine derivatives’ excretion (*p =* 0.132), and PDC index (*p =* 0.282) (Table 4).

The estimated purine derivatives’ excretion, using a spot urine sample collected at 4 h after the morning feeding, presented a negative linear effect with different levels of peach palm meal (*p =* 0.003), indicating that the urine output obtained with different average values of daily creatinine excretion observed in each diet increases the accuracy of the technique (Table 5).

## 4. Discussion

Although the diets were balanced to be isonitrogenous and isoenergetic, the decrease of dry matter intake (DMI) prompted a linear reduction in the intake of crude protein (CP), total digestible nutrients (TDN), and metabolizable energy (ME) by the inclusion levels of peach palm meal in the diets. In view of the results described for DMI, there was higher acceptability by the animals of the diet without peach palm meal, indicating that there was feed selection with the rejection of the peach palm meal by the lambs. It is possible that the increase in the total unsaturated fatty acid contents with the replacement of maize by peach palm meal affected the palatability, due to possible rancidification of the peach palm meal [5,6,47] and the presence of phenolic compounds [16,48,49].

The rejection caused food restriction which is associated with increased NPY protein levels in the hypothalamus [8]. Sugino et al. [9] reported that ghrelin, secreted mainly by the stomach, is a peptide that appears to participate in energy homeostasis by stimulating GH secretion and controlling feeding behavior. Circulating ghrelin levels have been shown to rise before a meal and fall afterward, suggesting that anticipation of a meal may stimulate secretion. Thus, ghrelin may play an important role in controlling feeding behavior and energy homeostasis. Foradori et al. [15] showed support for the hypothesis that during short-term fasting, systemic ghrelin concentrations and NPY expression in the arcuate nucleus rise. In the brain, NPY levels and NPY receptor density are changed in response to alterations in energy balance [15,22,50]. Anukulkitch et al. [50] reported data showing that diet-induced reduction in body weight leads to increased NPY expression in sheep.

Neuropeptide Y (NPY) is a co-transmitter of the sympathetic nervous system including the renal nerves. The kidney expresses NPY receptors, which can also be activated by peptide YY (PYY), a postprandial circulating hormone released from gastrointestinal cells [16,22,23,51]. Despite the profound reductions of renal blood flow, systemic NPY infusion can cause diuresis and natriuresis; this occurs largely independently of the pressure natriuresis mechanisms and is possibly mediated by an extrarenal Y5 receptor. NPY produces potent renal vasoconstriction via theY_1_ receptor, thus, the glomerular filtration rate may be affected [21,22,52].

It is known that protein intake leads to renal hyper-perfusion and plasma hyperfiltration in the renal glomerulus, so that it can enhance urine formation [25,53]. The possibility of higher plasma filtration in the kidneys of animals after feeding and protein intake can explain the filtered tubular load of creatinine and PD, thus increasing their urinary excretion [24,53,54,55,56].

As products of metabolism excreted in the urine, the concentrations of plasma creatinine and purine derivatives are determined by renal perfusion and filtration fraction in the kidneys, and variations in both have been observed in livestock [53]. In addition, Kiani et al. [56] showed that the concentration of plasma creatinine increased in adult ruminants fed a low protein diet.

The endogenous creatinine clearance can be used to measure the glomerular filtration rate, which is an indicator of kidney function [24,25]. Hence, the decrease of creatinine clearance could indicate that the renal activity in the lambs was affected by the diets with peach palm meal. Consistently, there was an increase in the concentration of plasma creatinine, possibly as a consequence of its lower clearance in response to the levels of peach palm meal in diets.

An increase in the concentrations of plasma creatinine was observed in adult ruminants, indicating that there are mechanisms enabling animals with a lower food intake to change the glomerular filtration rate [56]. The adaptation of the renal perfusion potentially may have an impact on concentrations of plasma creatinine, which does not effectively suffer tubular absorption and secretion [24].

Creatinine, whose excretion is considered as a fixed proportion relative to the metabolic body weight (muscular mass) and dietary factors, requires adjusting for variation in the urine or plasma volume [26,57,58]. However, Braun et al. [59] related that plasma creatinine is not an early indicator of kidney function, and Kiani et al. [56] observed that creatinine concentration in plasma is increased when renal perfusion and filtration fraction decrease in small ruminants fed a diet with low CP. Skotnicka et al. [53] reported that feeding time and diet (particularly the high protein content) can modify renal activity, which results in plasma hyperfiltration in renal glomeruli. 

In this present study, there was higher plasma creatinine in lambs that ingested lower amounts of CP and ME as a consequence of lower voluntary intake. On the other hand, the relationship of these phenomena to decreased creatinine excretion in sheep fed a diet with 40% peach palm meal level is worth investigating. However, Santos et al. [5] related better feed conversion at the 40% level of peach palm meal that could be associated with the increased energy density of the diet with the use of peach palm meal. The implications of this for adequately understanding the effects of food restriction and energy balance on the renal activity to obtain the purine derivatives excretion should not be overlooked.

Zoccali et al. [60] and Ezzat et al. [23] showed that greater serum creatinine was associated with increased serum NPY and decreased glomerular filtration rate. Moreover, increased voluntary intake and energy balance correlated with decreased NPY levels [50,61]. This is not surprising considering that the food restriction decreases the renal blood flow by direct mechanisms of intrarenal action of NPY, causing a decrease in the creatinine clearance. Diuresis may also occur secondarily by systemic hemodynamic changes as a consequence of multiple extrarenal effects of NPY, promoting increased filtration fraction. This would explain the increase in creatinine excretion, as was observed at levels of peach palm meal above 40% maize replacement [21,22,52].

The purine derivatives to creatinine ratio depends on the level of feed intake and intestinal flow of microbial purines and can indicate the intestinal flow of microbial nitrogen [31,62]. According to Chen et al. [30], the PDC index can be used as a marker to estimate the PDE because it considers the animal’s metabolic body weight.

In this present study, for the PDC index, there was imprecision in differentiating the diets, as was observed with the 24-h total urine sample and also when using the spot urine sample at a single time point at 4 h after morning feeding. Biased measurements were obtained with spot samples taken on a number of occasions during the 24-h cycle since there was significant linear and cubic variation. This may be explained by changed creatinine clearance in lambs fed a total mixed ration with peach palm meal. Hence, the PDC index was not effective in predicting purine derivatives’ excretion obtained in that urine spot samples collected at a single time point (4 h after morning feeding) and in urine obtained by the 24-h total urine collection.

Similarly, Nsahlai et al. [63] observed that diets had no effect on the purine derivatives to creatinine ratio, however, related that the poorest quality diet (teff straw) exhibited the lowest PDE in the daily total urine sample without affecting voluntary intake. Nevertheless, George et al. [62] reported that the purine derivatives to creatinine ratio was constant over a wide range of voluntary DMI at different times after feeding in crossbred bulls. The authors concluded that the spot urine sampling technique to predict the microbial protein supply is not suitable for detecting small differences in microbial nitrogen supply. Hence, the determination of PDE in total urine (mmol d^−1^) is necessary to assess precisely the microbial nitrogen supply.

In this study, there was a similar plasma concentration of purine derivatives in lambs that ingested lower amounts of both CP and ME. The urinary purine derivatives excretion ranged from 0.70 to 0.89 mmol kg BW^−0.75^ d^−1^, decreasing with increased levels of peach palm meal. Thus, as shown above, the total digestible nutrients’ intake decreased with the increased levels of peach palm meal that have probably contributed to causing a reduction in microbial protein synthesis, as reported by Santos et al. [64]. In the study of Santos et al. [64] the microbial protein synthesis was obtained in thirty lambs by spot urine sampling at 4 h after the morning feeding and different mean values of creatinine excretion for each diet as markers of urine output. Therefore, urinary PD excretion reflects microbial protein synthesis due to the high correlation between the two parameters [26,27,33,62,65].

Comparing the urine volume and purine derivatives’ excretion, there were no differences in the mean values obtained by spot sampling at different time points compared to the 24-h total urine. However, a single average value of daily creatinine excretion (28.12 mg kg^−1^ BW) obtained from experimental diets to calculate the urinary volume, was not suitable to assess accurately the purine derivatives’ excretion in lambs. The cubic response indicates interference from renal activity.

As there was no difference in the circadian rhythm of the PDE and PDC index, these were tested in the spot urine sample obtained at 4 h after the morning feeding, as suggested by [25,33,34]. Muszczyñski et al. [25] did not find a modifying effect of food (feeding time) on diurnal renal activity and reported a lack of GFR rhythms in goats with permanent food access. The results showed that the spot urine collection at 4 h was suitable for detecting differences between the experimental diets when different mean values of DCE were used (30.5, 27.9, 25.9, 27.8, and 28.4 mg kg^−1^ BW) from each diet. It indicates that a sole average value of daily creatinine excretion could not be used to estimate urine volume in spot urine samples, because the different levels of peach palm meal affected the creatinine clearance, indicating renal activity changed by diet. Aside from diet, the difference has to be underlined between ruminants’ species. In fact, urinary purine derivatives’ excretion (mmol kg BW^−0.75^ d) in the present study was intermediate [66] while the PDC index was lower [67] in lambs compared to cows and buffaloes. In addition, according to Thanh et al. [66], spot sampling is not reliable in buffaloes due to infrequent urination.

Similarly, Pereira et al. [68] compared urine sampling techniques (24-h total and spot urine) in lambs fed unconventional feedstuff, and suggested sampling at any time point after the feeding to obtain the PDE with the urine volume estimated by average daily creatinine excretion measured in each experimental diet.

Chen et al. [69] found that the purine derivatives to creatinine ratio in spot samples showed no significant difference when obtained at 1-h intervals of collection and it correlated with the purine derivatives’ excretion. It is consistent with the PDC index and purine derivatives excretions which were not affected by the different time points of spot urine collection. In contrast, in this present study, both showed a significant cubic component of diet effect suggesting interference with renal activity. Possibly, this could be associated with peach palm meal levels affecting the food intake. The lack of an effect of the collection time on the PDC index indicates the possibility of using a spot urine sample to obtain the PD excretion at any time point of the day.

However, the PDC index obtained, using spot urine at 4 h after morning feeding, was inaccurate in detecting dietary differences, indicating poor utility in this regard. If the creatinine clearance is affected by the diet, both the PDC index and also the urinary volume estimated by a single average value for daily creatinine excretion per body weight, do not allow a precise estimation of PD excretion.

Our hypothesis “the use of PDC index and a single average value of daily creatinine excretion as a marker of the urinary output is not suitable for estimating the purine derivatives’ excretion in a spot urine sample when diets change the renal activity” was not rejected. In addition, the PDC index was not precise in detecting the difference between levels of peach palm meal in diets. The spot urine sampling at 4 h is suitable in sheep for detecting differences in purine derivatives’ excretion [33] when the feed does not change the renal activity; otherwise, this technique requires validation by the 24-h total urine collection.

Results showed that peach palm meal in diets for lambs reduces the endogenous creatinine clearance, indicating the necessity to perform 24-h total urine collection in at least one individual providing the DCE for each diet to assess precisely the purine derivatives’ excretion in the spot urine of lambs.

## 5. Conclusions

Peach palm meal in diets decreases the endogenous creatinine clearance in lambs. A single average value for daily creatinine excretion and purine derivatives to creatinine index is inadequate as markers to estimate the urinary purine derivatives’ excretion in spot urine samples when the renal function is affected by feed. Therefore, it is recommended to use measured creatinine excretion for each diet. These findings represent a baseline for future studies on the interaction between low palatability compounds from unconventional foodstuffs, feed intake behavior, and renal function.

## Figures and Tables

**Table 1 animals-12-01195-t001:** Ingredient composition (g kg^−1^ DM) of the diets; chemical composition of the Tifton 85 hay; peach palm meal and experimental diets.

				Replacing Level, % of DM				
Ingredient compostion				0	10	40	60	85
Tifton 85 hay				300	300	300	300	300
Maize meal				508	457	298	204	77
Peach palm meal				0	51	210	304	431
Soybean meal				177	177	177	177	177
Mineral salt ^1^				15	15	15	15	15
Chemical composition		Tifton 85 hay	Peach palm meal	Experimental Diets				
				0	10	40	60	85
DM		921	926	929	925	928	922	927
OM		920	968	935	931	935	932	935
Ash		80	32	65	69	65	68	65
CP		55	80	135	135	139	137	149
NDIP		308	116	316	325	295	313	268
ADIP		229	201	206	200	212	209	179
EE		20	136	42	37	50	54	65
TC		839	752	752	746	734	740	725
NFC		24	624	288	317	327	368	366
NDF		762	116	464	430	407	372	359
ADF		470	68	224	224	214	233	231
LIG		103	10	33	33	34	33	41
TDN ^a^		477	809	763	710	778	749	779
ME, MJ kg^−1^ DM ^b^		--	9.6	11.79	11.68	12.36	12.81	12.65

DM, g/kg of natural matter; OM: Organic matter; CP: Crude protein; NDIP: Neutral detergent insoluble protein, g/kg of CP; ADIP: Acid detergent insoluble protein, g kg^−1^ of CP; EE: Ether extract; TC: Total carbohydrates; NFC: Non-fiber carbohydrates; NDF: Neutral detergent fiber, free of ash and protein; ADF: Acid detergent fiber; LIG: Lignin; TDN: Total digestible nutrients; ME: Metabolizable energy.^1^ Mineral salts (DM basis) 168 g kg^−1^ of Ca; 85 g kg^−1^ of P; 600 mg kg^−1^ of Cu; 1850 mg kg^−1^ of Fe; 45 mg kg^−1^ of Co; 80 mg kg^−1^ of I; 1350 mg kg^−1^ of Mn. ^a^ TDN calculated according to Weiss (1999); ^b^ ME estimated by equation (NRC, 2001).

**Table 2 animals-12-01195-t002:** Nutrients’ intake in lambs fed diets (D) containing peach palm meal replacing maize.

Item	Replacing Level, % of DM					SEM	*p*-Value		
	0	10	40	60	85		D	L	Q
g d^−1^									
DMI ^1^	858.0	752.0	666.0	576.0	538.0	47.7	0.029	0.002 ^a^	0.601
CPI ^2^	120.0	102.0	94.0	82.2	84.0	6.65	0.059	0.006 ^b^	0.288
EEI ^3^	22.52	23.34	38.22	22.82	37.79	2.56	0.805	0.807	0.627
TDNI ^4^	626.0	556.0	514.0	454.0	416.0	0.04	0.098	0.008 ^c^	0.819
MJ d^−1^									
MEI ^5^	9.90	7.05	8.69	4.08	6.76	0.59	0.125	0.012 ^d^	0.786
g kg BW^−0.75^									
DMI ^1^	79.0	68.4	62.5	51.3	52.1	3.36	0.007	0.005 ^e^	0.346
CPI ^2^	14.47	8.40	8.32	4.78	8.06	0.51	0.063	0.010 ^f^	0.245
EEI ^3^	2.80	2.34	3.80	2.68	3.72	0.19	0.633	0.478	0.492
TDNI ^4^	58.04	50.42	48.04	40.58	40.36	0.05	0.042	0.004 ^g^	0.526
MJ kg BW^−0.75^									
MEI ^5^	1.23	0.71	0.87	0.48	0.65	0.04	0.084	0.010 ^h^	0.540

DM dry matter; MJ Mega joule; BW Body weight; ^1^ Dry matter intake; ^2^ Crude protein intake; ^3^ Ether extract intake; ^4^ Total digestible nutrients intake; ^5^ Metabolizable energy intake. ^a^ Ŷ = 802.7 − 3.200 X; ^b^ Ŷ = 111.9 − 0.397 X; ^c^ Ŷ = 630.1 − 2.220 X; ^d^ Ŷ = 8.76 − 0.0376 X; ^e^ Ŷ = 73.7 − 0.0307 X; ^f^ Ŷ = 11.41 − 0.0668 X; ^g^ Ŷ = 54.77 − 0.0193 X; ^h^ Ŷ = 1.01 − 0.0056 X; X = peach palm meal levels.

**Table 3 animals-12-01195-t003:** Measured urine volume; endogenous creatinine clearance (CC); plasma and urinary creatinine; plasma and urinary purine derivatives (PD); PD/creatinine index using four days of 24-h total urine collections (C); and plasma concentration in lambs fed diets (D) containing peach palm replacing maize.

Item	Replacing Level, % of DM					SEM	*p*-Value			Length of Urine Collection				*p*-Value	
	0	10	40	60	85		D	Linear	Quadratic	24 h (Day 1)	48 h (Day 2)	72 h (Day 3)	96 h (Day 4)	C	D × C
Urine, L d^−1^	0.90	0.82	0.55	1.06	0.86	0.10	0.008	0.614	0.008 ^a^	0.87	0.82	0.83	0.83	0.980	0.990
CC ^1^															
ml min^−1^	70.8	56.2	45.9	47.4	38.7	2.18	<0.0001	<0.0001 ^b^	0.071	53.8	59.4	46.9	46.8	0.032 ^c^	0.984
mL kg BW^−0.75^ min^−1^	6.66	5.37	4.33	4.31	3.64	0.20	<0.0001	<0.0001 ^d^	0.099	4.99	5.55	4.45	4.48	0.038 ^e^	0.968
Creatinine															
plasma, mg dl^−1^	0.72	0.78	0.89	0.97	1.30	0.07	0.095	0.011 ^f^	0.358						
urine, mg kg BW^−1^	30.5	27.9	25.9	27.9	28.4	0.49	0.016	0.125	0.004 ^g^	27.8	29.4	27.9	27.3	0.280	0.990
mmol kg BW^−0.75^ d^−1^	0.59	0.54	0.50	0.54	0.55	0.01	0.011	0.150	0.003 ^h^	0.54	0.57	0.54	0.53	0.260	0.990
Purine derivatives															
plasma, mmol L^−1^	1.84	1.89	1.63	1.36	1.89	0.19	0.818	0.718	0.515						
urine, mmol d^−1^	9.79	9.50	7.76	8.13	7.20	0.31	<0.0001	<0.0001 ^i^	0.562	8.28	8.55	8.58	8.51	0.919	0.986
mmol kg BW^−0.75^ d^−1^	0.89	0.88	0.73	0.72	0.70	0.02	<0.0001	<0.0001 ^j^	0.304	0.77	0.79	0.79	0.78	0.971	0.972
PDC index ^1^	16.9	18.8	16.5	15.6	14.2	0.75	0.139	0.034	0.271	15.3	15.2	17.1	17.9	0.241	0.873

^1^ PD, mmol L^−1^ ÷ creatinine, mmol L^−1^ × kg BW ^0.75^; BW = body weight. ^a^ Y = 0.92 (±0.09117, *p <* 0.0001) − 0.01630 (±0.004227, *p =* 0.0002) X + 0.000189 (±0.000049, *p =* 0.0002) X^2^; ^b^ Y = 59.1 (±2.4386, *p <* 0.0001) − 0.2543 (±0.05491, *p <* 0.0001) X; ^c^ contrast, 48 h vs. (24 h + 72 h + 96 h) (*p* < 0.01); ^d^ Y = 5.70 (±0.2560, *p <* 0.0001) − 0.02595 (±0.005254, *p <* 0.0001) X; ^e^ contrast, 48 h vs. (24 h + 72 h + 96 h) (*p* < 0.01); ^f^ Ŷ = 0.73 (±0.048, *p <* 0.0001) + 0.004557 (±0.001591, *p =* 0.009) X; ^g^ Y = 29.9 (±0.9117, *p <* 0.0001) − 0.1587 (±0.05298, *p =* 0.004) X + 0.001691 (±0.000588, *p =* 0.005) X^2^; ^h^ Y = 0.57 (±0.001695, *p <* 0.0001) − 0.00303 (±0.001019, *p =* 0.004) X + 0.000034 (±0.000012, *p =* 0.005) X^2^; ^i^ Y = 9.54 (±0.5111, *p <* 0.0001) − 0.02801 (±0.007868, *p =* 0.001) X; ^j^ Y = 0.88 (±0.03193, *p <* 0.0001) − 0.00247 (±0.000611, *p =* 0.0001) X; X = peach palm meal levels.

**Table 4 animals-12-01195-t004:** Estimated urine volume, excretion of purine derivatives (PD) and PD/creatinine index using the first day (day 1) of 24-h total urine collection and urine spot samples collected at 4-h intervals of time (T) after morning feeding in lambs fed diets (D) containing peach palm meal replacing maize.

Item	Replacing Level, % of DM					SEM	*p*-Value	Time of Urine Collection, h						Day1	SEM	*p*-Value	
	0	10	40	60	85		D	4	8	12	16	20	24			T	D × T
Urine^1^, L d^−1^	1.30	0.85	0.71	1.43	1.08	0.11	0.009 ^a^	1.11	1.08	1.13	1.35	0.88	0.88	0.87	0.07	0.314	0.911
Purine derivatives ^1^																	
mmol d^−1^	9.51	12.52	9.96	8.05	8.37	0.61	0.0001 ^b^	9.73	8.39	9.29	10.91	9.36	10.41	8.28	0.34	0.132	0.762
mmolkgBW^−0.75^ d^−1^	0.88	1.16	0.93	0.74	0.80	0.05	0.0002 ^c^	0.94	0.77	0.87	0.99	0.87	0.96	0.77	0.03	0.138	0.805
PDC index ^2^	18.0	22.4	17.6	14.3	16.3	1.40	0.004 ^d^	19.9	15.3	16.5	19.5	16.6	18.4	16.2	0.68	0.282	0.937

^1^ Estimated by single mean value of daily creatinine excretion = 28.12 mg kg^−1^ BW (body weight); ^2^ PD, mmol L^−1^ ÷ creatinine, mmol L^−1^ × kg BW ^0.75^. ^a^ Polynomial contrast for cubic component (*p* = 0.007); ^b^ Polynomial contrast for linear (*p* = 0.003) and cubic (*p* = 0.0007) component; Y = 10.3030 (±0.5483, *p* < 0.0001) − 0.02671 (±0.01054, *p* = 0.012) X; ^c^ Polynomial contrast for linear (*p* = 0.006) and cubic (*p* = 0.0003) component; Y = 0.9489 (±0.0456, *p* < 0.0001) − 0.00225 (±0.000923, *p* = 0.016) X; ^d^ Polynomial contrast for linear (*p* = 0.015) and cubic (*p* = 0.002) component; Y = 18.6607 (±0.9120, *p* < 0.0001) − 0.05063 (±0.01983, *p* = 0.015) X; X = peach palm meal.

**Table 5 animals-12-01195-t005:** Urinary purine derivatives (PD) and PD/creatinine index using spot samples collected at 4 h after morning feeding in lambs fed diets (D) containing peach palm replacing maize.

Item	Replacing Level, % of DM					SEM	*p*-Value		
	0	10	40	60	85		D	Linear	Quadratic
Purine Derivatives ^1^									
mmol d^−1^	13.1	11.9	9.5	7.8	6.3	0.87	0.054	0.004 ^a^	0.987
mmol kg BW^−0.75^ d^−1^	1.29	1.12	0.90	0.72	0.62	0.09	0.038	0.003 ^b^	0.734
PDC index ^2^	21.3	24.9	16.7	13.4	23.3	0.68	0.636	0.698	0.445

^1^ Estimated by following values of daily creatinine excretion: 30.5, 27.9, 25.9, 27.8, and 28.4 mg kg^−1^ BW (body weight) obtained in each respective diet; ^2^ PD, mmol L^−1^ ÷ creatinine, mmol L^−1^ × kg BW ^0.75^.^a^ Ŷ = 13.059 (±0.6767, *p <* 0.0001) − 0.08375 (±0.01413, *p <* 0.0001) X; ^b^ Ŷ = 1.228 (±0.1347, *p <* 0.0001) − 0.00783 (±0.002354, *p =* 0.003) X; X = peach palm meal levels.

## Data Availability

The data used to generate the results in the paper are not available.

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
