# Peer review of "Creatinine as a Urinary Marker of the Purine Derivatives Excretion in Urine Spot Samples of Lambs Fed Peach Palm Meal"

_animals, 2022, doi:10.3390/ani12091195_

Round 1
Reviewer 1 Report
This paper reports results from a trial comparing diferent levels of peach palm meal on purine derivative excretion in lambs.
The main goal of the trial seems to be purine derivative excretion, creatinitn excretion in urine and the possibility to assess them with a single sample of urine.
So, the comparision of diets is considered as a mean to study that with a diet that decreases the intake.
This is not clear in all the manuscript, it could be improved (for example in conclusion).
At l 204, VU = the urine volum (ml) per minute and at l 210 VU = estimated volume urine (L). Could you precise the difference ?or use 2 different appellations if needed.
The presentation of results is not clear from l 294 to l 310. Could you include a graph to show the difference obtained with a spot at 4h and estimated and measured values for some parameters and different diets? It is not easy to compare them in the different tables.
In the discussion, the beginning is devoted to general information not linked to the results you have obtained (l 323 - l 391). It is quite long et a part of this could be moved to the introduction.
L332 do you have some information about rancidity in your diet ?
A graphic summary could facilitate the understanding of the manuscript.
In table 1, do you have no value for ME in tifton 85 hay ?
Author Response
Response to Reviewer 1 Comments
This paper reports results from a trial comparing diferent levels of peach palm meal on purine derivative excretion in lambs. The main goal of the trial seems to be purine derivative excretion, creatinine excretion in urine and the possibility to assess them with a single sample of urine.
Point 1: So, the comparision of diets is considered as a mean to study that with a diet that decreases the intake. This is not clear in all the manuscript; it could be improved (for example in conclusion).
Response: Thank you for your comments. Yes, we concluded the manuscript clarifying that there are two factors that can cause fluctuations in the concentration of creatinine and consequently the metabolites (derived from purine) in the urine. One is the volume of urine produced, which is dependent on water intake, where, in the calculation of excretion, the volume of urine corrects for this change in metabolite levels. The other factor would be the change in the glomerular filtration rate, which does not necessarily translate into a change in urinary volume. If the diet promotes a reduction in the glomerular filtration rate, there may be a decrease in the concentration of metabolites in the urine, including the daily excretion of creatinine. Therefore, a single mean value for the daily excretion of creatinine and purine derivatives for the creatinine index is inadequate as markers for estimating urinary excretion of purine derivatives in point urine samples when renal function is affected by diet.
Point 2: At l 204, VU = the urine volum (ml) per minute and at l 210 VU = estimated volume urine (L). Could you precise the difference? or use 2 different appellations if needed.
Response: Thank you very much for your question. The creatinine clearance is done to evaluate the function of the kidneys, which is done from the comparison of the concentration of creatinine in the blood and eliminated in the urine during 24 hours. In this way, the result indicates the amount of creatinine that was taken from the blood and eliminated in the urine, and as this process is carried out by the kidneys, changes in the results may be indicative of kidney damage.
The calculation of endogenous creatinine clearance can be done using different formulas, the most common being that of Pitts (1974), which takes into account the 24-hour total creatinine concentration in the urine (mg dl−1), the plasma creatinine concentration (mg dl−1) and the volume of urine (ml) per minute. Therefore, the presentation of the urine volume expressed in (ml) is a requirement of the formula to obtain the endogenous creatinine clearance, a situation different from the presentation of the diary urine volumes of the animals.
Point 3: The presentation of results is not clear from l 294 to l 310. Could you include a graph to show the difference obtained with a spot at 4h and estimated and measured values for some parameters and different diets? It is not easy to compare them in the different tables.
Response: Thank you very much for your suggestion. However, we believe that our work is a complete article, and that this form of presentation in tables is the most detailed form of the data (the numbers vary between the different sources of information), allowing a better interpretation by the readers. Thus, we believe that our results are new insights into the excretion of purine derivatives in spot urine samples from lambs.
Point 4: In the discussion, the beginning is devoted to general information not linked to the results you have obtained (l 323 - l 391). It is quite long et a part of this could be moved to the introduction.
Response: Thank you, the instructions have been revised.
Point 5: L332 do you have some information about rancidity in your diet?
Response: Thank you very much for your question. Yes, the lipid rancidity of peach palm meal is a probable factor affecting intake by rejection. This was clearly noticeable when checking the orts from the trough of the diets during the present experiment; as the level of substitution of maize for peach palm meal was elevated, the animals rejected the concentrate. It is possible that the increase in the EE contents with the substitution of maize for the peach palm meal affected the palatability, due to a possible rancidification that occurred as the peach palm meal replaced the maize.
Point 6: A graphic summary could facilitate the understanding of the manuscript.
Response: Thank you very much for your suggestion. Done.
Point 7: In table 1, do you have no value for ME in tifton 85 hay?
Response: Yes. We are sorry for absence of this data.

Reviewer 2 Report
See the attached file

Author Response
Response to Reviewer 2 Comments
Creatinine as a urinary marker of the purine derivatives excretion in urine spot samples of lambs fed peach palm meal
By replacing maize with different percentage of peach palm, the authors verified the influence of diet
on creatinine clearance with the aim to demonstrate the unsuitability of the PDC index and of a single value of daily creatinine excretion as markers of the urinary output for estimating the purine derivatives excretion in a spot urine sample. The authors used adequate methodologies and well depicted and discussed their work. In my opinion, the paper needs only few corrections and/or additionm as follows:
Point 1: Line 112-113: the authors indicate as 150 gkg-1 and 11.72 MJ kg-1 on a dry matter the average values of diets which were actually different, as reported in Table 1. Please clarify.
Response: Thank you for your comments. You can find these changes at line (94-95) of the revised manuscript.
Point 2: Line 428: in order to improve the discussion, I suggest to add this sentence: “Aside from diet, it has to be underlined the difference among ruminant’s species. In fact, urinary purine derivatives excretion (mmol kg BW-0.75 d) in present study was intermediate (Cutrignelli et al., 2007) while PDC index was lower (Thanh et al., 2004) in lambs compared to cow and buffaloes. In addition, according to Thanh et al. (2004) spot sampling is not reliable in buffaloes due to infrequent urination”.
Response: Thank you for your suggestions. You can find these recommendations at line (431-435) of the revised manuscript.
Point 3: References – please add the following two.
Response: Thank you for your suggestions. You can find these recommendations at line (651-656) of the revised manuscript.
M.I. Cutrignelli, G. Piccolo, S. D’Urso, S. Calabrò, F. Bovera, R. Tudisco, F. Infascelli (2007). Urinary excretion of purine derivatives in dry buffalo and Fresian cows, Italian Journal of Animal Science, 6:sup2, 563-566,
VO THI KIM THANH, DAO THI PHUONG, TRAN THI THU HONG, PHUNG THI, LUU, NGO MAU DUNG, HOANG QUOC HUNG. E.R. ØRSKOV (2004) Comparison of purine derivatives and creatinine in plasma and urine between local cattle adn buffaloes in Vietnam. In: H.P.S. Makkar and X.B. Chen (eds.), Estimation of Microbial Protein Supply in Ruminants Using Urinary Purine
Derivatives, 75–85. © 2004 Kluwer Academic Publishers.
